# Antagonistic role of the BTB-zinc finger transcription factors Chinmo and Broad-Complex in the juvenile/pupal transition and in growth control

**Sílvia Chafino[1,2,3], Panagiotis Giannios[1,3], Jordi Casanova[1,3], David Martín[2]\*, Xavier Franch-Marro[2]\***

[1]Institut de Biologia Molecular de Barcelona (CSIC), Barcelona, Spain; [2]Institute of Evolutionary Biology (IBE, CSIC-Universitat Pompeu Fabra), Barcelona, Spain; [3]Barcelona Institute of Science and Technology, Institute for Research in Biomedicine, IRB Barcelona, Barcelona, Spain

**\*For correspondence:**
david.martin@ibe.upf-csic.es
(DM);
xavier.franch@ibe.upf-csic.es
(XF-M)

**Competing interest:** The authors declare that no competing interests exist.

**Abstract** During development, the growing organism transits through a series of temporally regulated morphological stages to generate the adult form. In humans, for example, development progresses from childhood through to puberty and then to adulthood, when sexual maturity is attained. Similarly, in holometabolous insects, immature juveniles transit to the adult form through an intermediate pupal stage when larval tissues are eliminated and the imaginal progenitor cells form the adult structures. The identity of the larval, pupal, and adult stages depends on the sequential expression of the transcription factors *chinmo*, *Br-C,* and *E93*. However, how these transcription factors determine temporal identity in developing tissues is poorly understood. Here, we report on the role of the larval specifier *chinmo* in larval and adult progenitor cells during fly development. Interestingly, *chinmo* promotes growth in larval and imaginal tissues in a Br-C-independent and -dependent manner, respectively. In addition, we found that the absence of *chinmo* during metamorphosis is critical for proper adult differentiation. Importantly, we also provide evidence that, in contrast to the well-known role of *chinmo* as a pro-oncogene, Br-C and E93 act as tumour suppressors. Finally, we reveal that the function of *chinmo* as a juvenile specifier is conserved in hemimetabolous insects as its homolog has a similar role in *Blatella germanica*. Taken together, our results suggest that the sequential expression of the transcription factors Chinmo, Br-C and E93 during larva, pupa an adult respectively, coordinate the formation of the different organs that constitute the adult organism.

## Editor's evaluation

This important study demonstrates that the transcription factor Chinmo is a master regulator that maintains larval growth and development as part of the metamorphic gene network in *Drosophila*. Chinmo does so in part by regulating Broad expression in imaginal tissues (e.g. eye and wing discs) and in a Broad-independent manner in other larval tissues such as the salivary glands and larval trachea. Finally, the authors demonstrate that the role of Chinmo in promoting larval development is conserved between holometabolous insects and hemimetabolous insects, which lack a pupal stage. The data were collected and analyzed using solid and validated methodology and will be of interest to a broad audience including those interested in development and evolution.

**eLife digest** Egg, larva, pupa, adult: the life of many insects is structured around these four well-defined stages of development. After hatching, the larva grows until it reaches a certain size; when the right conditions are met, it then becomes a pupa and metamorphoses into an adult. Most larval cells die during metamorphosis; only a group known as imaginal cells survives, dividing and maturing to create pupal and adult tissues.

Each of these developmental steps are linked to a particular genetic program deployed in response to a single stage-specifying gene. For instance, the activation of the *Br-C* gene triggers the transition from larva to pupa, while *E93* initiates the transformation of the pupa into an adult. However, which stage-specifying gene controls larval identity remains unclear. Recent studies suggest that in fruit flies, a gene known as *chinmo* could be playing this role.

In response, Chafino et al. explored how *chinmo* shapes the development of fruit fly larvae. The experiments showed that *chinmo* is activated in the juvenile stage, and that it is required for the larvae to grow properly and for larval and imaginal tissues to form. Conversely, it must be switched off for the insect to become a pupa and then an adult. Further work suggested that the role of *chinmo* as a larval specifier could have emerged early in insect evolution.

Moreover, Chafino et al. revealed that *chinmo* could repress *Br-C*, an important characteristic since stage-specifying genes usually switch on sequentially by regulating each other. A closer look suggested that, in imaginal cells, *chinmo* promotes development by inhibiting *Br-C*; in larval cells, however, *chinmo* not only has a *Brc*-repressing role but it is also necessary for larval cells to grow. Additional experiments exploring the role of the stage-specifying genes in tumor formation showed that *chinmo* promotes cells proliferation while *Br-C* and *E93* had tumor-suppressing properties.

Overall, the work by Chafino et al. sheds new light on the genetic control of insect development, while also potentially providing a new perspective on how genes related to *chinmo* and *Br-C* contribute to the emergence of human cancers.

## Introduction

Animal development passes through various stages characterised by distinct morphological and molecular changes. In humans, for instance, development continues from birth through to childhood and puberty to give rise to the adult form. As in many animals, in holometabolous insects such as *Drosophila melanogaster*, the developmental stages are sharply defined: embryogenesis gives rise to the larva, a juvenile stage, which, upon different rounds of growth and moulting, brings about a new stage structure, the pupa, when most of the larval cells die and the adult progenitor cells (imaginal cells) develop to generate the adult organism. The regulation of stage-specific differences is mediated by the action of two major developmental hormones, the steroid 20-hydroxyecdysone and the terpenoid juvenile hormone (**Hiruma and Kaneko, 2013**; **Jindra et al., 2013**; **Truman, 2019**; **Truman and Riddiford, 2007**; **Truman and Riddiford, 2002**; **Yamanaka et al., 2013**). Both hormones exert this precise developmental control by regulating the expression of three critical genes that encode for the stage-identity factors that compose the metamorphic gene network: the C2H2 zinc finger type factor *Krüppel-homolog 1* (*Kr-h1*), the helix-turn-helix *Ecdysone inducible protein 93*F (*E93*), and *Broad-complex* (*Br-C*; also known as broad), a member of the bric-a-brac-tramtrack-broad family (**Martín et al., 2021**).

The deployment of the pupal-specific genetic program is controlled by the expression of *Br-C* at the larval-pupal transition (**Truman, 2019**; **Zhou and Riddiford, 2002**). Upon the formation of the pupa, hormone signalling triggers the expression of the helix-turn-helix factor *E93*, whose product represses *Br-C* expression and directs the formation of the final differentiated adult structures (**Chafino et al., 2019**; **Martín et al., 2021**; **Ureña et al., 2014**). While it is firmly established that *Br-C* and *E93* are the stage-specifying genes for the pupal and adult states, the nature of the larval specifying gene has been elusive. To date, larval identity has been attributed to Kr-h1, which is present during the larval period and represses *Br-C* and *E93* expression during this period (**Huang et al., 2011**; **Ureña et al., 2016**). However, although Kr-h1 is undoubtedly critical for maintaining the larval state, evidence has shown that this factor cannot be considered the larval specifier per se. For example, depletion of *Kr-h1* in *Drosophila* does not prevent normal larval development nor a timely transition to the pupa

(*Beck et al., 2004*; *Pecasse et al., 2000*). In this regard, the product of *chronologically inappropriate morphogenesis* (*chinmo*) gene, another member of the BTB family of transcription factors, has been recently proposed to be responsible for larval identity in *Drosophila* (*Truman and Riddiford, 2022*).

First isolated based on its requirement for the temporal identity of mushroom body neurons (*Zhu et al., 2006*), the identification of Chinmo as a more general larval specifier has provided invaluable insights into the molecular mechanisms underlying the control of juvenile identity. Yet, little is known about how this factor exerts its function along with Br-C and E93. Moreover, given that holometabolous insects are comprised of both larval tissues and pools of adult progenitor cells (known as imaginal cells), a central issue in the understanding of how larval identity is controlled is how larval and imaginal cells respond differentially to the same set of temporal transcription factors. Furthermore, in the sequential activation of *chinmo*, *Br-C,* and *E93*, the extent of the activity directly attributable to each transcription factor or to their mutual repression is still unclear.

Here, we confirm the role of *chinmo* as larval specifier in larval and imaginal cells and establish its regulatory interactions with the other temporal specifiers. We also examine how the temporal sequence of Chinmo and Br-C differently affects with the genetic program that establishes larval vs. imaginal identity. Thus, we found that Chinmo controls larval development of larval and imaginal tissues in a Br-C-independent and -dependent manner, respectively. According to these data, and in the context of the metamorphic gene network, we also show that *chinmo* absence is critical for the transition from larva to pupa and then to adult, as it acts as a repressor of both *Br-C* and *E93*. In addition, we report that the *chinmo* homologue has a similar role in the cockroach *Blattella germanica,* thereby indicating that its function as a juvenile specifier precedes the hemimetabolous/holometabolous split. Finally, we show that in contrast to the well-characterised role of *chinmo* as a pro-oncogene, the *Br-C* pupal and *E93* adult specifiers act mainly as tumour suppressor genes. These characteristics are maintained beyond insects and may account for the different role of some human BTB-zinc finger transcription factors in tumourigenesis.

## Results and discussion

### *chinmo* is expressed throughout larval stages and is required in larval and imaginal tissues

Examination of *chinmo* expression revealed that it is expressed during embryogenesis and early larval development and that it is strongly downregulated from L3 (*Figure 1A*). Immunostaining analysis in imaginal and larval tissues confirmed the presence of Chinmo in L1 and L2 stages and its disappearance in late L3 (*Figure 1B and C*), an expression profile that is in agreement with previous studies (*Narbonne-Reveau and Maurange, 2019*; *Truman and Riddiford, 2022*). We next addressed its functional requirement by knocking down this factor with an RNAi transgene controlled by the ubiquitous *Act-Gal4* driver. *chinmo*-depleted animals showed developmental arrest at the end of the first instar larval stage presenting a tanned cuticle clearly reminiscent of the tanned larval cuticle of the puparium (*Figure 1D*). Consistent with the phenotype, we found that arrested *chinmo*-depleted larvae precociously expressed pupal cuticle genes while blocked larval-specific genes activation (*Figure 1E*). These results confirm that *chinmo* is required for normal progression of the organism during the larval period, as proposed by *Truman and Riddiford, 2022*.

Since *Drosophila* larva consists of a combination of larval and imaginal tissues, we then analysed the contribution of *chinmo* to the development of these two types of tissues. Regarding the former, *chinmo* was selectively depleted in the salivary glands using the *forkhead* (*fkh*) driver (*fkh-Gal4*), which is active in this tissue from embryogenesis onwards. The salivary glands are a secretory organ that develops from embryonic epithelial placodes (*Abrams et al., 2003*; *Bradley et al., 2001*; *Edgar et al., 2014*; *Zielke et al., 2013*). This tissue is responsible for producing glycosylated mucin for the lubrication of food during the larval period (*Costantino et al., 2008*; *Farkaš et al., 2014*; *Riddiford, 1993*; *Syed et al., 2008*) and for synthesising glue proteins for the attachment of the pupa to a solid surface at the onset of metamorphosis (*Andres et al., 1993*; *Costantino et al., 2008*; *Kaieda et al., 2017*). As it is shown in *Figure 2A*, although depletion of *chinmo* in the salivary glands did not affect the formation of this organ, it caused a dramatic decrease in normal larval development, as revealed by the strong reduction in size and DNA content of the gland cells (*Figure 2B–D*). Consistently, the expression levels of both early and late specific salivary gland protein encoding genes, such as *new*

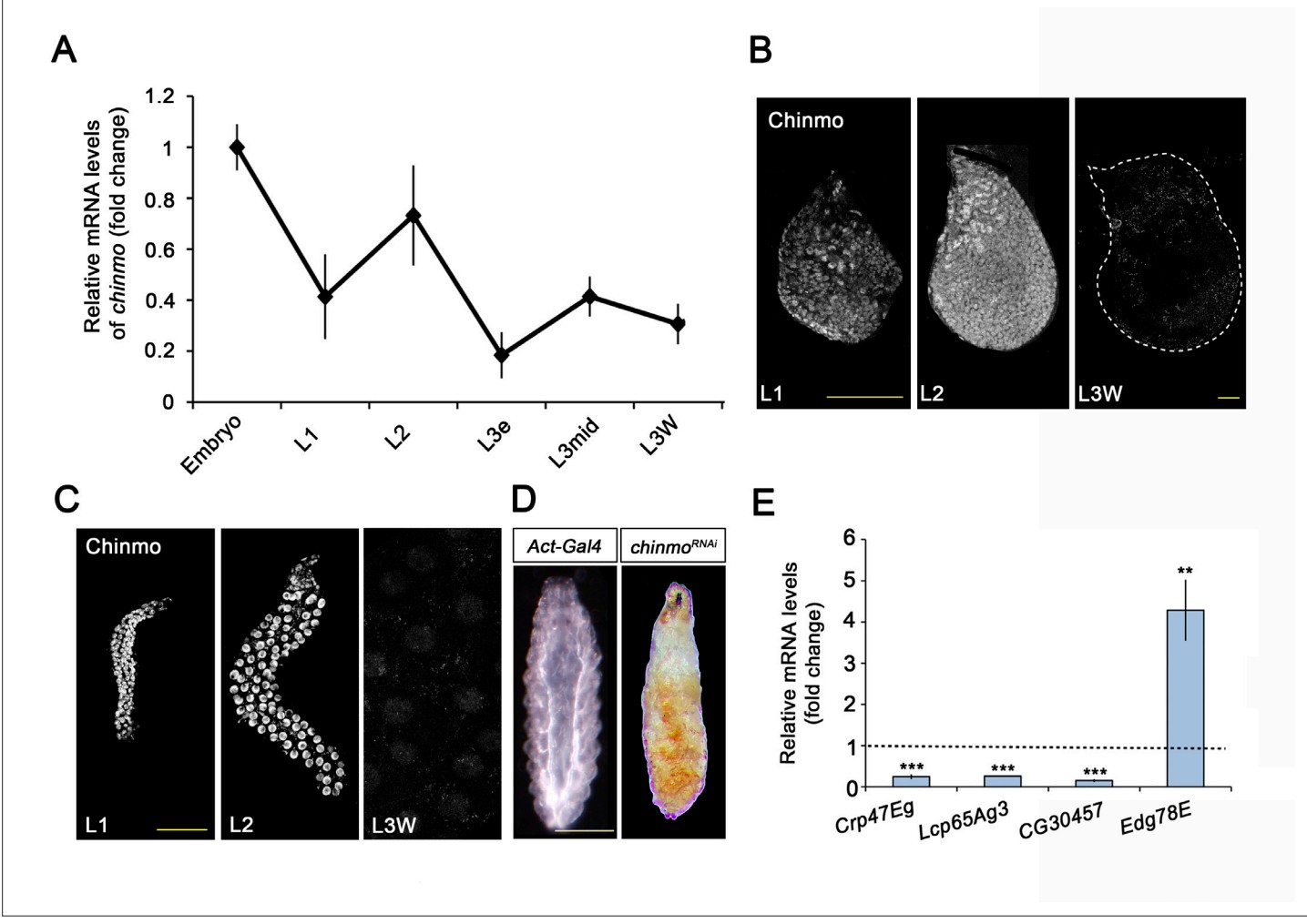

**Figure 1.** Chinmo is expressed during early larval stages and is essential for proper larval development. (**A**) *chinmo* mRNA levels measured by quantitative real-time reverse transcriptase polymerase chain reaction (qRT-PCR) from embryo to the wandering stage of L3 (L3W). Transcript abundance values were normalised against the *Rpl32* transcript. Fold changes were relative to the expression of embryo, arbitrarily set to 1. Error bars indicate the SEM (n = 3). (**B–C**) Chinmo protein levels in the wing disc (**B**) and salivary glands (**C**) of larval L1, L2, and L3W (females) stages. (**D**) Compared with the control (*Act-Gal4*), overexpression of *UAS chinmo*^RNAi in the whole body induced developmental arrest at the L1 stage. Scale bars represent 50 μm (**B and C**) and 0.5 mm (**D**). (**E**) Relative expression of larval-specific (*Crp47Eg, Lcp65Ag3,* and *CG30457*) and pupal-specific genes (*Edg78E*) in *UAS-chinmo*^RNAi L1 larvae measured by qRT-PCR. Transcript abundance values were normalised against the *Rpl32* transcript. Fold changes were relative to the expression in control larvae, arbitrarily set to 1 (dashed black line). Error bars indicate the SEM (n = 3). Statistical significance was calculated using t test (***p ≤ 0.001; **p ≤ 0.005).

The online version of this article includes the following source data for figure 1:

**Source data 1.** Numerical data for *Figure 1A and E*.

glue 1–3 (ng) and *Salivary gland secretion* (*Sgs*), were virtually undetectable in *chinmo*-depleted salivary glands compared to control (*Figure 2E*). Remarkably, to further study the requirement of *chinmo* for larval tissue growth, we analyzed the role of this factor in the larval tracheal system. Although depletion of *chinmo* specifically in the tracheal cells, using a *trh-Gal4* driver, resulted in many arrested L2 larvae with a necrotic tracheal system, escapers that reached L3 presented reduction in nuclear size and DNA content of tracheal cells as well as reduced length of the organ (*Figure 2—figure supplement 1*), thus confirming that Chinmo is required for proper growth of larval tissues.

Regarding the role of *chinmo* in imaginal tissues, we knocked down this factor in the pouch region of wing imaginal discs from the embryonic period onwards using the *escargot* (*esg*) driver (*esg-Gal4*). As before, depletion of *chinmo* in the *esg* domain did not alter the specification of the disc, but strongly impeded its larval development. Thus, in late L3 wing discs only the notum, which does not express

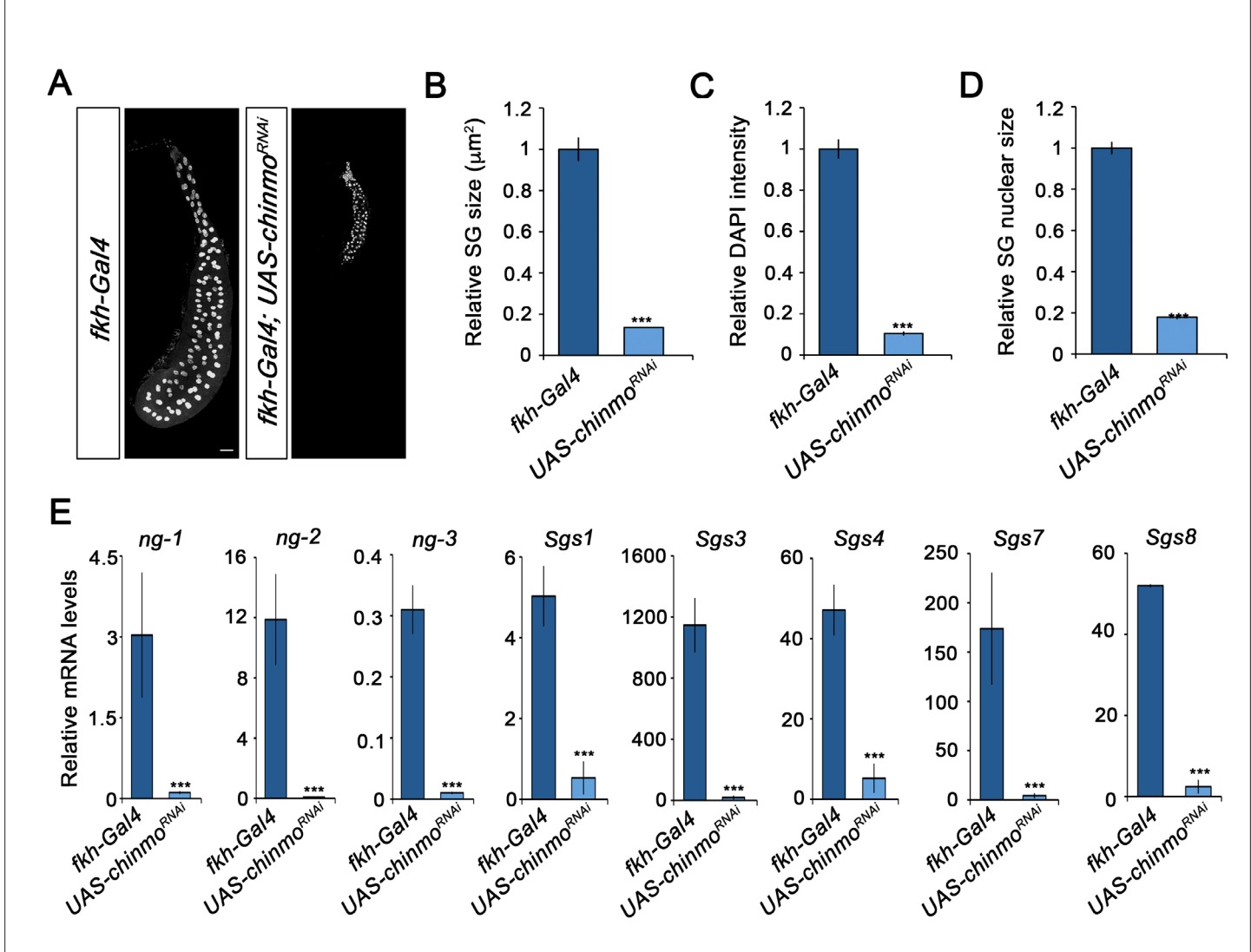

**Figure 2.** Chinmo is required for proper growth and function of the salivary glands during larval development. (**A**) DAPI staining of salivary glands from control (*fkh-Gal4*) and *UAS-chinmo^RNAi* larvae at L3W. Scale bar represents 50 μm. (**B–D**) Comparison of the relative size of salivary glands (n = 10 for each genotype) (**B**), DAPI intensity (n = 50 for each genotype) (**C**), and nucleic size of salivary glands (n = 50 for each genotype) (**D**) between *UAS-chinmo^RNAi* and control larvae at L3W. Error bars indicate the SEM (n = 5–8). (**E**) Relative expression of *ng1-3* and *Salivary glands secretion* genes (*Sgs*) in *UAS-chinmo^RNAi* L3W animals measured by quantitative real-time reverse transcriptase polymerase chain reaction (qRT-PCR). Transcript abundance values were normalised against the *Rpl32* transcript. Error bars indicate the SEM (n = 5–8). Statistical significance was calculated using t test (***p≤0.001).

The online version of this article includes the following source data and figure supplement(s) for figure 2:

**Source data 1.** Numerical data for *Figure 2B–E*.

**Figure supplement 1.** The role of *Chinmo* in the larval tracheal system.

**Figure supplement 1—source data 1.** Numerical data for *Figure 2—figure supplement 1D-F*.

the *esg-Gal4* driver, was observed while the wing pouch, revealed by positive GFP signal, was strongly reduced and did not show the expression of patterning genes such as *wingless* (*wg*) and *cut* (*ct*) (*Figure 3A*). In line with these results, although most of the *chinmo*-depleted animals arrested development as pharate adults, escapers that were able to eclose (15%) had no wings (*Figure 3—figure supplement 1*). Similarly, depletion of *chinmo* in the eye disc using a specific driver (*ey-Gal4*) induced similar effects abolishing the developing tissue and the formation of the adult eye (*Figure 3—figure supplement 2*). Taken together, these data show that Chinmo is required during the larval period to control the development and function of larval and imaginal tissues.

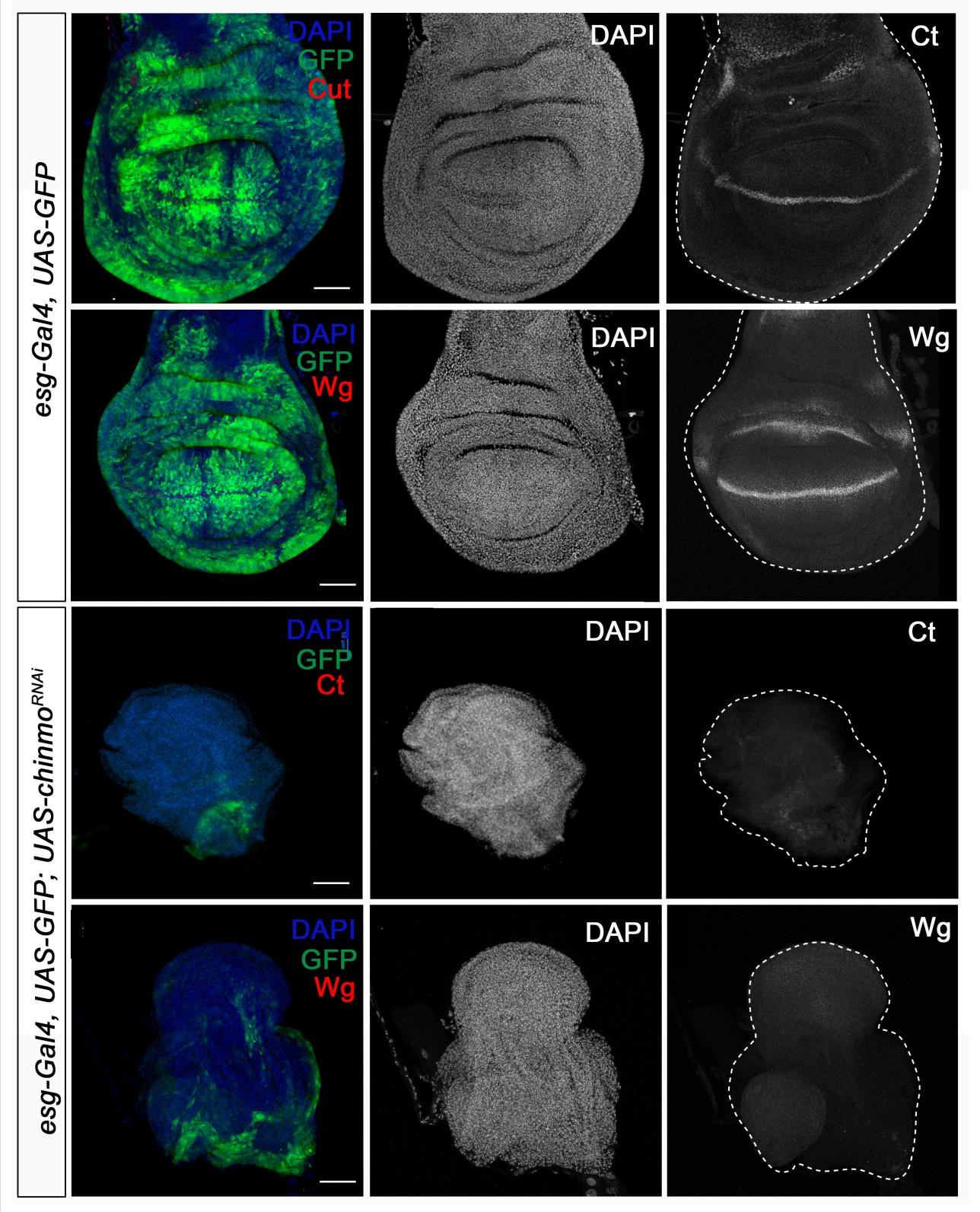

**Figure 3.** Chinmo is necessary for wing development during the larval period. Expression of Ct and Wg in wing discs of control (*esg-Gal4*) and *UAS-chinmo^RNAi* L3W larvae. Wing discs were labelled to visualise the *esg* domain (GFP in green) and nuclei (DAPI). Ct and Wg were not detected in *UAS-chinmo^RNAi*. Scale bars represent 50μm.

*Figure 3 continued on next page*

**Figure supplement 1.** Chinmo is required for wing development during the larval period.

**Figure supplement 2.** *Br-C*-dependent requirement of *chinmo* in the eye disc.

## Distinct roles of Chinmo in larval and progenitor cells

A critical feature of the metamorphic gene network factors is that their sequential expression is achieved through a series of regulatory interactions between them. Therefore, we next sought to characterise the regulatory interactions of Chinmo with the pupal specifier Br-C and the adult specifier E93. To this end, we measured the expression of *Br-C* and *E93* in *chinmo*-depleted salivary glands and wing discs. Contrary to recently published data (***Truman and Riddiford, 2022***), both tissues showed a significant and premature increase of Br-C protein levels as early as in L1 larvae, while no increase in E93 protein levels was detected in any tissue (***Figure 4***).

In view of these results, we speculated whether the impairment of larval development observed in *chinmo*-depleted animals could be the result of precocious presence of the wrong stage-identity factor, in this case, Br-C. To address this notion, we precociously expressed Br-CZ1, the main Br-C isoform expressed during imaginal larval development (***Narbonne-Reveau and Maurange, 2019***), in salivary glands and wing discs. As previously described, ectopic expression of Br-CZ1 blocked Chinmo activation (***Narbonne-Reveau and Maurange, 2019***). As a consequence, precocious upregulation of Br-C blocked development in both tissues, phenocopying the loss of function of *chinmo* (***Figure 4—figure supplement 1***). This result suggests that a fundamental function of Chinmo is to suppress the expression of the pupal specifier *Br-C* during the juvenile stages. To confirm this hypothesis, we simultaneously depleted *chinmo* and *Br-C* in salivary glands and wing discs. Remarkably,

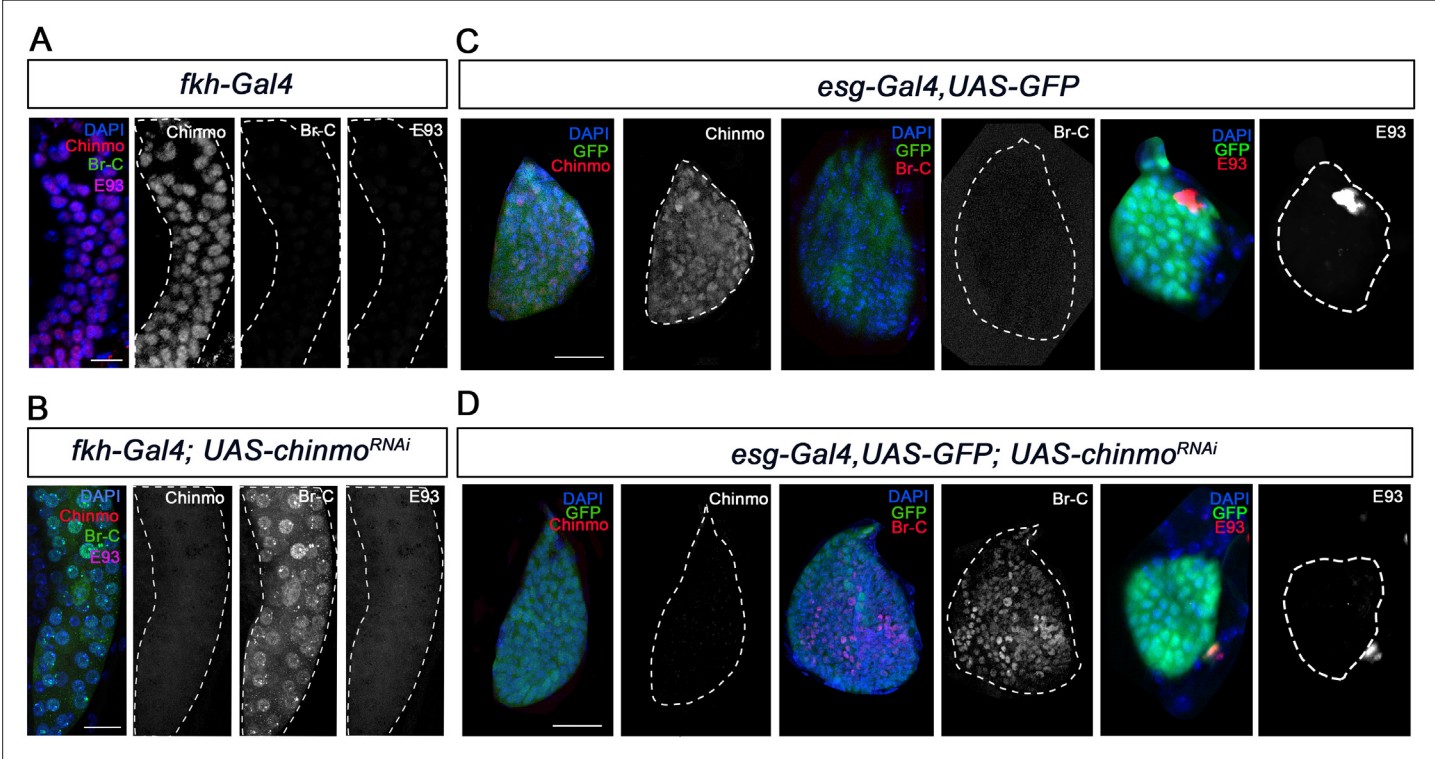

**Figure 4.** Chinmo represses Br-C in salivary glands and wing discs during early larval development. (**A–B**) Expression of Chinmo, Br-C, and E93 in salivary glands of L1 control (*fkh-Gal4*) (**A**), and *UAS-chinmo^RNAi* (**B**). (**C–D**) Expression of Chinmo, Br-C, and E93 in wing discs of early L2 control (*esg-Gal4*) (**C**) and *UAS-chinmo^RNAi* (**D**). The *esg* domain is marked with GFP and all cell nucleus with DAPI. In the absence of *chinmo* only *Br-C* shows early upregulation in both tissues. Scale bars represent 25 μm.

The online version of this article includes the following figure supplement(s) for figure 4:

**Figure supplement 1.** Overexpression of Br-CZ1 phenocopies *chinmo* loss of function in SGs and wing discs.

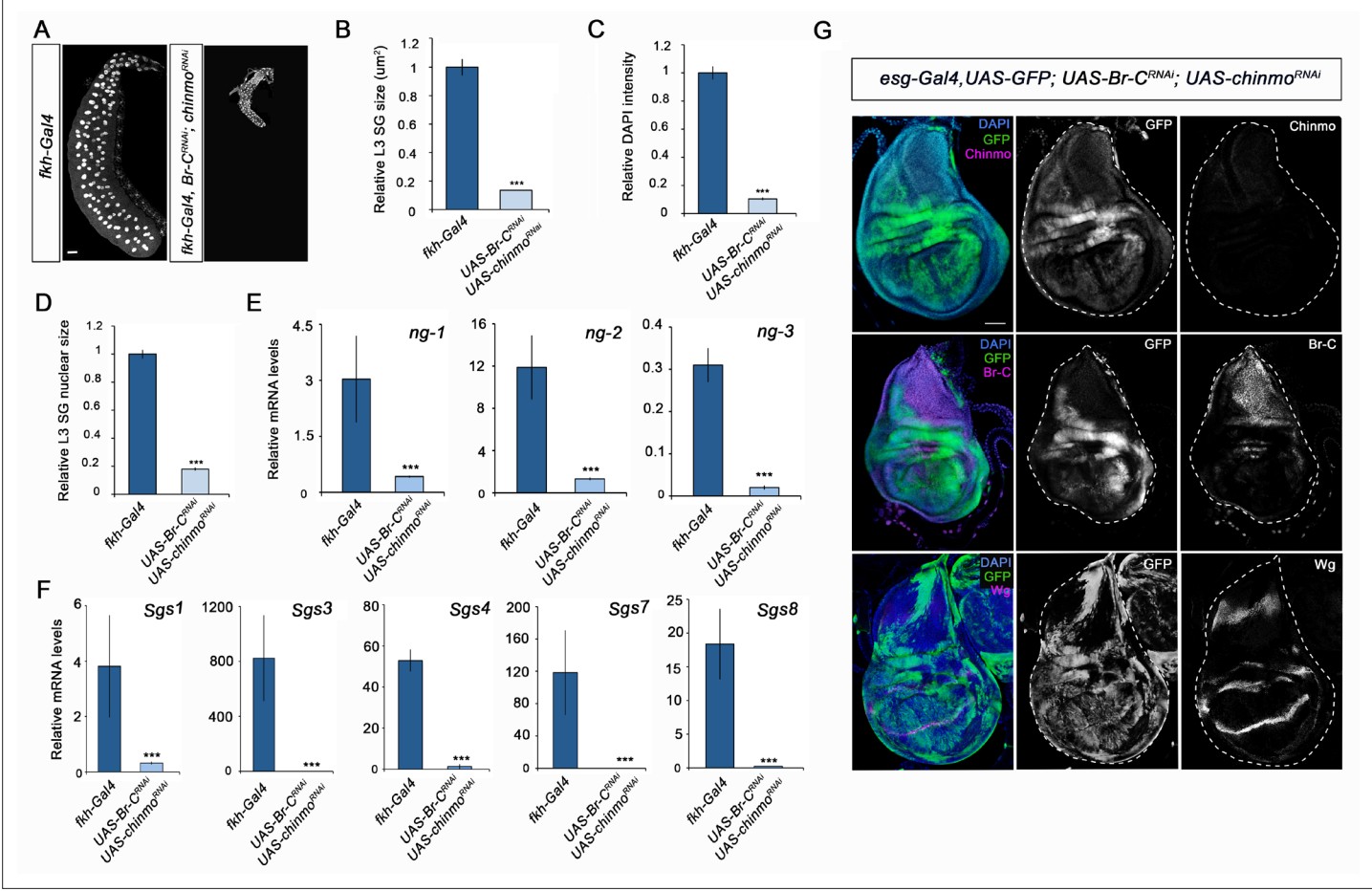

**Figure 5.** Different requirement of *chinmo* for the larval growth of salivary glands and wing discs. (**A**) DAPI-stained salivary glands from control (*fkh-Gal4*) and *UAS-Br-C^RNAi*; *UAS-chinmo^RNAi* L3W larvae. In the absence of *chinmo* and Br-C, salivary glands did not grow. (**B–D**) Comparison of the relative size of salivary glands (n = 10 for each genotype) (**B**), DAPI intensity (n = 50 for each genotype) (**C**), and nucleic size of salivary glands (n = 30 for each genotype) (**D**) of control and *UAS-Br-C^RNAi*; *UAS-chinmo^RNAi* L3W larvae. (**E–F**) Relative expression of (**E**) *ng1-3* and (**F**) *Salivary glands secretion* genes in control and *UAS-Br-C^RNAi*; *UAS-chinmo^RNAi* L3W larvae measured by quantitative real-time reverse transcriptase polymerase chain reaction (qRT-PCR). Transcript abundance values were normalised against the *Rpl32* transcript. Error bars in B and C indicate the SEM (n = 5–8). Statistical significance was calculated using t test ( ****$p \leq 0.001$). (**G**) Expression of Chinmo, Br-C, and Wg in wing discs of *UAS-Br-C^RNAi*; *UAS-chinmo^RNAi* L3W larvae. Wing discs labelled to visualise the *esg* domain (GFP in green). In the absence of *chinmo* and *Br-C*, wing discs grow normally and express Wg correctly. Scale bars represent 50 μm.

The online version of this article includes the following source data and figure supplement(s) for figure 5:

**Source data 1.** Numerical data for *Figure 5B–F*.

**Figure supplement 1.** Effectiveness of *chinmo* and *Br-C* RNAis in the salivary glands.

**Figure supplement 2.** Expression of Chinmo and Br-C in the developing wing disc.

(**A–C**) Control wing discs, esg-Gal4; UAS-GFP, were labelled to visualise the esg domain in green and in red (**A**) Br-C expression in L3W larvae, (**B**) Chinmo expression in early-mid L3 larvae and (**C**) the morphogenetic marker Wg in L3W larvae. Scale bar represents 50 μm in all panels.

whereas salivary glands showed the same growth impairment observed upon *chinmo* depletion (*Figure 5A-F*, *Figure 5—figure supplement 1*), depletion of *Br-C* largely rescued the abnormalities in the wing discs caused by depletion of *chinmo*: the double knock-out wing discs developed in a regular manner to reach normal size by the end of L3 and showed proper expression of patterning genes such as *wg* (*Figure 5G*, *Figure 5—figure supplement 2*). The difference between larval and imaginal tissues was also observed in the analysis of the tracheal system and the eye imaginal disc. Whereas depletion of *Br-C* in the eye disc rescued the phenotype induced by the absence of *chinmo* (*Figure 3—figure supplement 1*), the larval trachea failed to restore the growth defects observed in *chinmo*-depleted tracheal cells (*Figure 2—figure supplement 1*). Taken together, our results

suggest that a major regulatory function of *chinmo* during early larval development in imaginal cells is channelled through the repression of *Br-C*, while in larval tissues *chinmo* appears to exert specific growth-related functions that are independent to *Br-C* repression. Thus, in the imaginal cells *chinmo* appears to ensure the expression of juvenile genes by repressing *Br-C*, a well-known inhibitor of larval gene expression (*Zhou and Riddiford, 2002*). In this regard, it is tempting to speculate that *Br-C* might repress the early expression of critical components of signaling pathways such as Wg and EGFR, involved in wing fate specification in early larval development (*Ng et al., 1996*; *Wang et al., 2000*; *Zecca and Struhl, 2002*). In contrast, in larval tissues *chinmo* seems to exert an active role promoting growth and maturation. The fact that the Br-C-dependent *Sgs* genes fail to be activated in absence of *chinmo*, when *Br-C* is prematurely expressed, supports this idea (*Figure 2E*). This different response could be explained by the nature of the larval and imaginal tissues. While larval tissues are mainly devoted to growth during the larval period and then fated to die during the metamorphic transition, the developmental identity of the imaginal cells is modified along the larva-pupa-adult temporal axis to give rise to the adult structures. This difference could also account for the distinct roles of the other members of the metamorphic gene network in larval and imaginal tissues. Thus, while *Br-C* is necessary for the degeneration of the larval salivary glands during the onset of the pupal period (*Jiang et al., 2000*), it is critical for the correct eversion of the imaginal wing disc and for the temporary G2 arrest that synchronises the cell cycle in the wing epithelium during early pupa wing elongation (*Guo et al., 2016*). Likewise, *E93* is necessary to activate autophagy for elimination of the larval mushroom body neuroblasts in late pupae (*Pahl et al., 2019*), whereas it controls the terminal adult differentiation of the imaginal wing during the same period (*Ureña et al., 2016*; *Uyehara et al., 2017*).

## Downregulation of *chinmo* is required during metamorphosis

The functional and expression data reported above show that Chinmo acts as a larval specifier in *Drosophila*. From this, we could infer that its absence by the end of larval development is required first for the transition to the prepupa, and then to allow terminal adult differentiation during the pupal period. If this were the case, maintenance of high levels of *chinmo* during late L3 would interfere with the larva-pupal transition. To test this possibility, we maintained high levels of *chinmo* in late L3 wing discs using the *Gal4/Gal80ts* system. Consistent with this hypothesis, overexpression of *chinmo* from early L3 in the anterior compartment of the disc using the *cubitus interruptus ci-Gal4* driver impaired its larva-pupal transition as abolished *Br-C* expression and induced apoptosis at late L3 as revealed by the high expression of the effector caspase Dcp-1 (*Figure 6A*). As a result, the size of the anterior compartment was dramatically reduced, and the expression of patterning genes such as *ct* was halted (*Figure 6B*). Impairment of *ct* expression was not just a consequence of cell death, as *ct* expression was neither detected in wing discs overexpressing both *chinmo* and the *p35* inhibitor of effector caspases (*Hay et al., 1994*; *Figure 6C and D*), but instead to a distinct response to the sustained expression of *chinmo* or to the consequent depletion of *Br-C*.

An alternative way to keep high levels of *chinmo* at late L3 is by depleting *Br-C*, a well-known repressor of *chinmo* from mid L3 (*Narbonne-Reveau and Maurange, 2019*). Therefore, we knocked down *Br-C* in the anterior compartment of the wing disc and confirmed that Chinmo levels remained high in this compartment by late L3. Also, in this case we observed a strong Dcp-1 staining and impairment of *ct* expression (*Figure 6E and F*). Importantly, simultaneous depletion of *chinmo* and *Br-C* from early L3 did not lead to an increase in apoptosis (*Figure 6G*) nor altered the expression of patterning genes (*Figure 5F*), which indicates that tissue death at the end of the larval period is due to sustained expression of *chinmo* rather than the absence of *Br-C*. Altogether, these results confirm that the transition from larva to pupa must take place in the absence of the larval specifier Chinmo.

Next, we analyzed whether lack of *chinmo* is also important during the pupal period to allow the E93-dependent development of the adult. To this end, we used the thermo-sensitive system to overexpress *chinmo* in the anterior part of the wing specifically during the pupal stage. To that aim, larvae were maintained at 18°C until 12 hr after pupa formation (APF) and then shifted to 29°C to allow the Gal4 to function. The resulting ectopic expression of *chinmo* led to a marked decrease in E93 protein levels (*Figure 7A*). As a result, the anterior compartment of the wing was strongly undifferentiated, a phenotype reminiscent of that observed in *E93*-depleted wings (*Ureña et al., 2016*; *Ureña et al., 2014*; *Figure 7B*). Taken together, our results show that *chinmo* must be downregulated during the

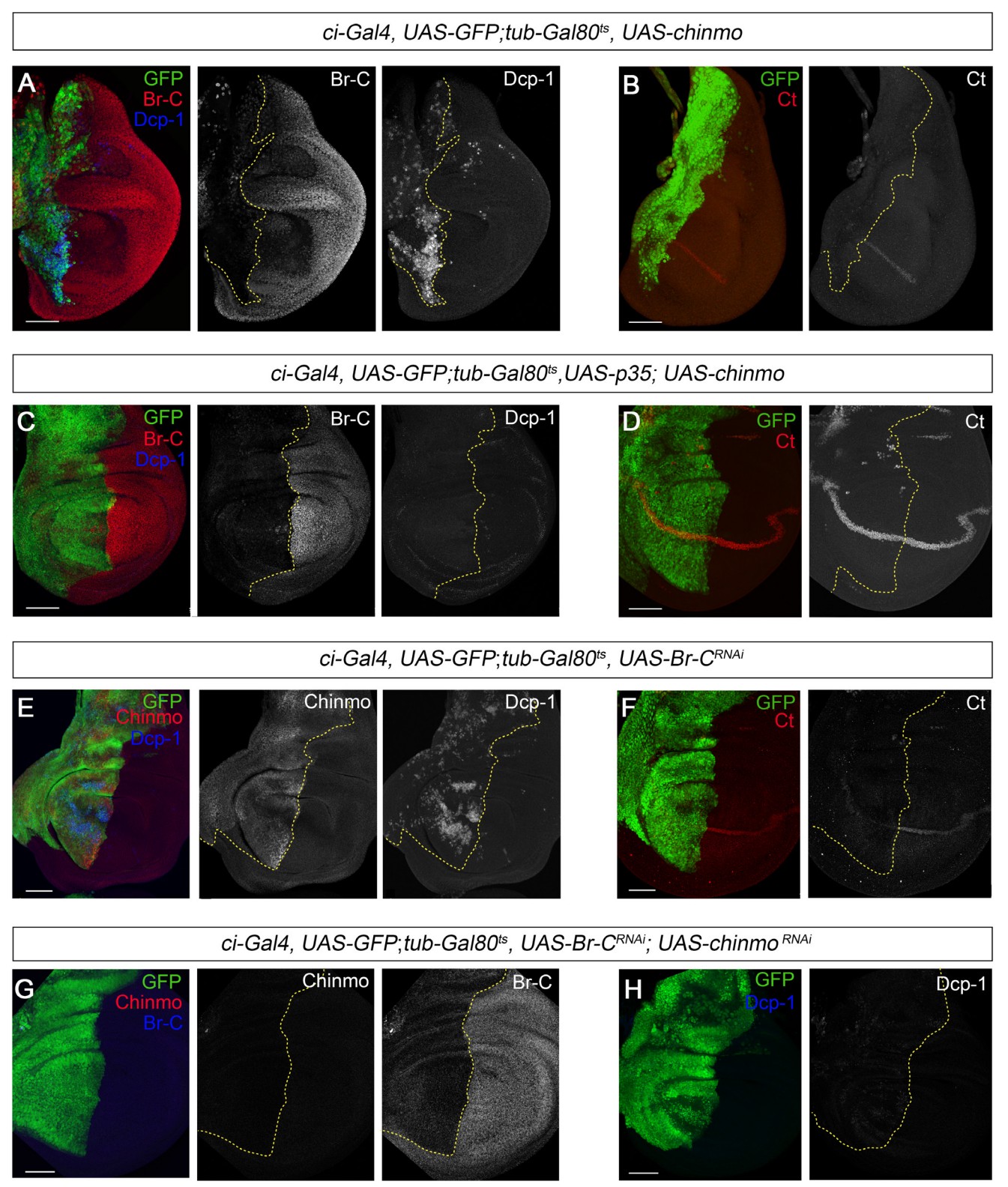

**Figure 6.** *chinmo* depletion during late L3 is required for proper larva to pupa transition. (**A–H**) Images of wing imaginal discs from L3W larvae. The indicated constructs were expressed under the control of the *ci-Gal4* driver. Overexpression or depletion of the transgenes was activated in early L3 larvae and analyzed at the L3W stage. An *UAS-GFP* construct was used to mark the anterior region of the disc where the transgenes were induced or repressed (green). (**A**) Overexpression of *chinmo* repressed Br-C, induced Dcp-1, and (**B**) abolished Ct. (**C**) Overexpression of *chinmo* together with *p35*

*Figure 6 continued on next page*

*Figure 6 continued*

repressed Br-C and blocked Dcp-1, but fails to restore normal expression of Ct (**D**). (**E**) Depletion of *Br-C* induced Chinmo and Dcp-1 and (**F**) repressed Ct. (**G**) In double depletion of *Br-C* and *chinmo* (**H**), Dcp-1 was not detected. Scale bars represent 50 μm.

initiation and throughout the metamorphic transition to allow the sequential expression of the pupal specifier *Br-C* and the adult specifier *E93*.

## Antagonistic effects of *chinmo* and *Br-C/E93* in tumour growth

Chinmo and Br-C belong to the extended family of BTB-zinc finger transcription factors, which are not restricted to insects. In humans, many such factors have been implicated in cancer, where they have opposing effects, from oncogenic to tumour suppressor functions (*Siggs and Beutler, 2012*). However, while overexpression of *Drosophila chinmo* has been found to cooperate with Ras or Notch to trigger massive tumour overgrowth (*Doggett et al., 2015*), changes in *Drosophila Br-C* expression have not been associated with any effect on tumourigenesis. Since the results described here, and those from other labs (*Narbonne-Reveau and Maurange, 2019*), indicate that *chinmo* and *Br-C* have antagonistic effects in terms of proliferation vs. differentiation, we addressed whether these opposite features might also be associated with pro-oncogenic or tumour suppressor properties, respectively. To test this notion, we resorted to the well-defined tumourigenesis model in *Drosophila* generated by the depletion of cell polarity genes such as *lgl* (*Froldi et al., 2008*; *Gong et al., 2021*). To downregulate *lgl* and generate an oncogenic sensitised background (*Figure 8A, B and G*), we triggered the expression of *UAS-lgl^{RNAi}* constructs in the imaginal wing disc pouch by means of *nubbin-Gal4* (*nub-Gal4*) (see Materials and methods for details).

Interestingly, RNAi-mediated depletion of *Br-C* in the wing discs in the downregulated *lgl* background resulted in an increase in the mean wing pouch volume compared to the downregulation of *lgl* alone (*Figure 8B, C and G*). Consistently, overexpression of *Br-C* in the same *lgl* background had the opposite effect, reducing the size of the *lgl*-induced overgrowth, thereby confirming that *Br-C* expression elicits tumour suppressor activity (*Figure 8D and G*). Given that *E93* has a similar pro-differentiation role to that of *Br-C*, we examined whether *E93* also exerts tumour suppressor activity. We found that overexpression of *E93* also reduced the size of *lgl* overgrowth (*Figure 8E and G*). However, as *E93* overexpression in the wing disc triggers extensive cell death (data not shown), we assessed whether in this case the reduction of the pouch region was caused by apoptosis induction. However, combined overexpression of *E93* with the *p35* inhibitor of apoptosis still lead to a reduction

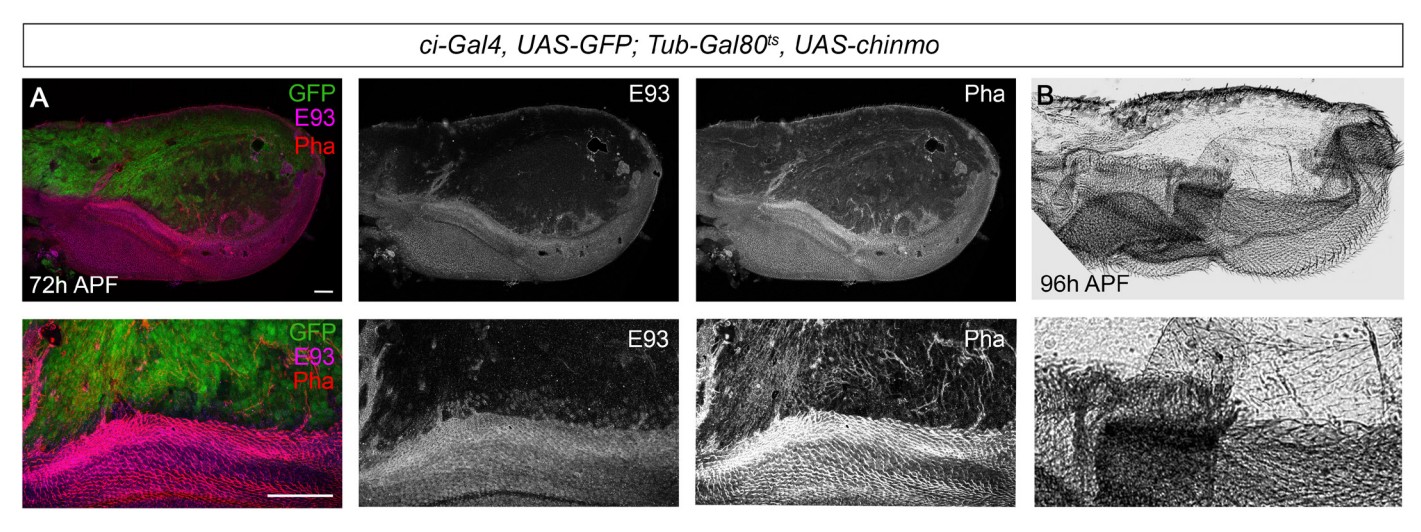

**Figure 7.** Presence of Chinmo during pupal development blocks adult differentiation. (**A**) Overexpression of *chinmo* in the anterior part of the pupal wing at 72 hr after pupa formation (APF) using *ci-Gal4* driver represses *E93* expression and produced alterations in phalloidin (Pha) pattern. (**B**) Cuticle preparation of a pupal wing at 96 hr APF expressing *chinmo* under the control of the *ci-Gal4* driver. Bottom panels are magnifications from upper images. The scale bars represent 50 μm (top panels) and 100 μm (bottom panels).

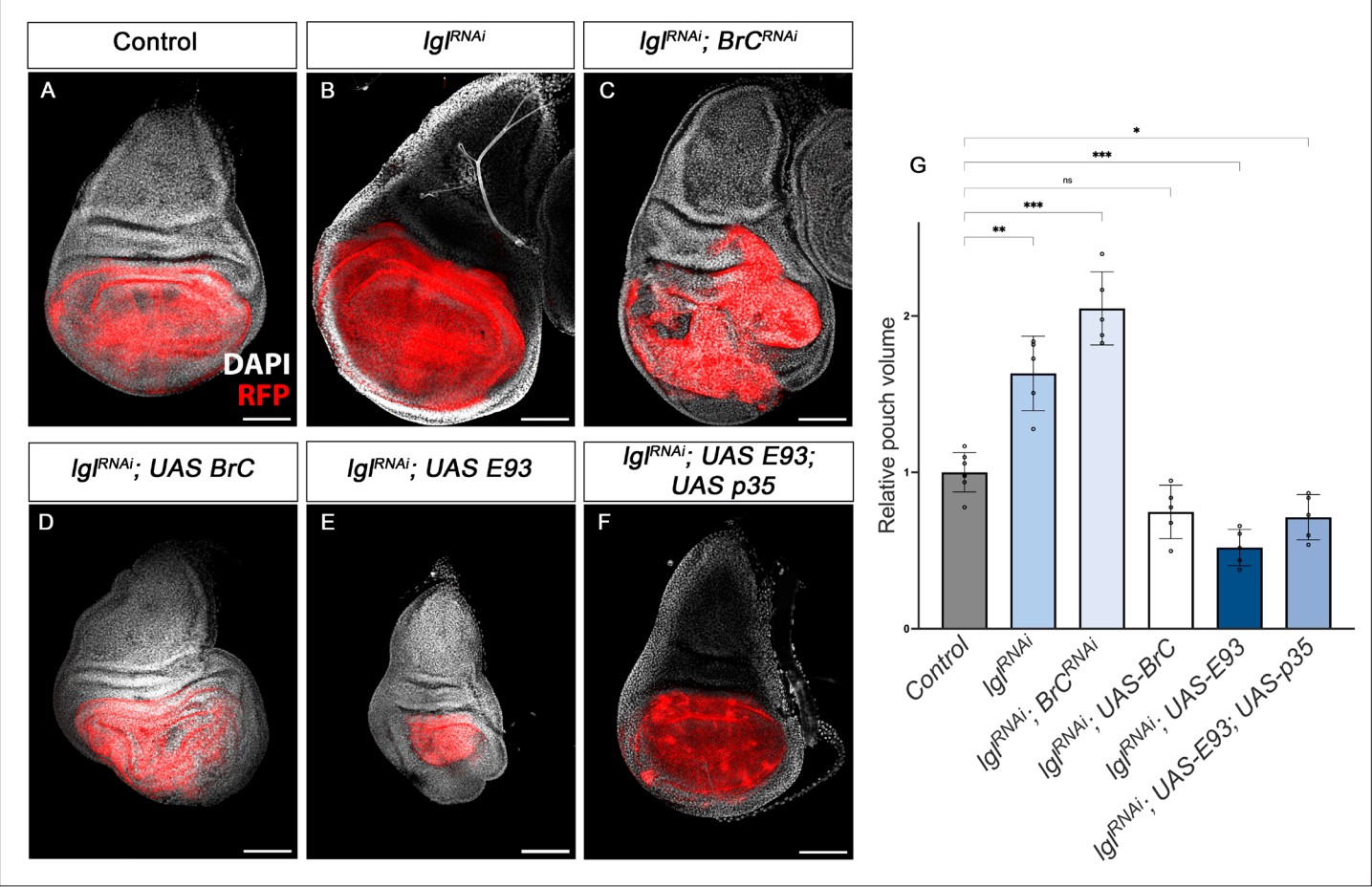

**Figure 8.** Tumour suppression action of Br-C and E93. (**A–F**) Confocal images of L3 wing imaginal discs. The indicated constructs were expressed under the control of the *nub-Gal4* driver. An *UAS-RFP* construct was used to mark the pouch region of the disc where the transgenes were induced (magenta). Nuclei were labelled with DAPI (grey). Scale bars at 100 µm. (**G**) Volumetric quantification of the RFP-positive area of the wing discs for the indicated groups. The pouch volumes were normalised to the mean of the control. Error bars in G indicate the SEM (n = 10). Welch's ANOVA (p<0.0001) followed by Dunnett's T3 post hoc tests (*p<0.05, **p<0.01, ***p<0.001).

The online version of this article includes the following source data for figure 8:

**Source data 1.** Numerical data for *Figure 8G*.

in the size of the wing pouch in the *lgl*-sensitised background (*Figure 8F and G*). Thus, also in this regard, *chinmo* and *Br-C/E93* play opposite functions, *chinmo* with pro-oncogenic features and *Br-C* and *E93* act as tumour suppressor genes.

## Role of *chinmo* in hemimetabolous development

As full metamorphosis is an evolutionary acquisition of holometabolous insects from hemimetabolous ancestors with no pupal stage (*Truman, 2019*), we sought to determine whether the role of *chinmo* as a larval specifier was also present in hemimetabolous insects. To this end, we used the German cockroach *B. germanica* as a model for hemimetabolous development. *Blattella* goes through six juvenile nymphal instars (N1–N6) before developing into an adult. Metamorphosis takes place during N6 and is restricted to the transformation of the wing primordia into functional wings, the attainment of functional genitalia, and changes in cuticle pigmentation (*Ureña et al., 2014*).

A detailed Tblastn search in the *Blattella* genome database revealed the presence of a *chinmo* orthologue (*Bg-chinmo*). To study *Bg-chinmo*, we first examined its expression during the life cycle of *Blattella*. We found that it is highly expressed in embryos and decreases dramatically thereafter during nymphal development (*Figure 9A*). In order to analyze the function of the relative low levels of *Bg-chinmo* during postembryonic stages, we analysed the function of *Bg-Chinmo* by systemic

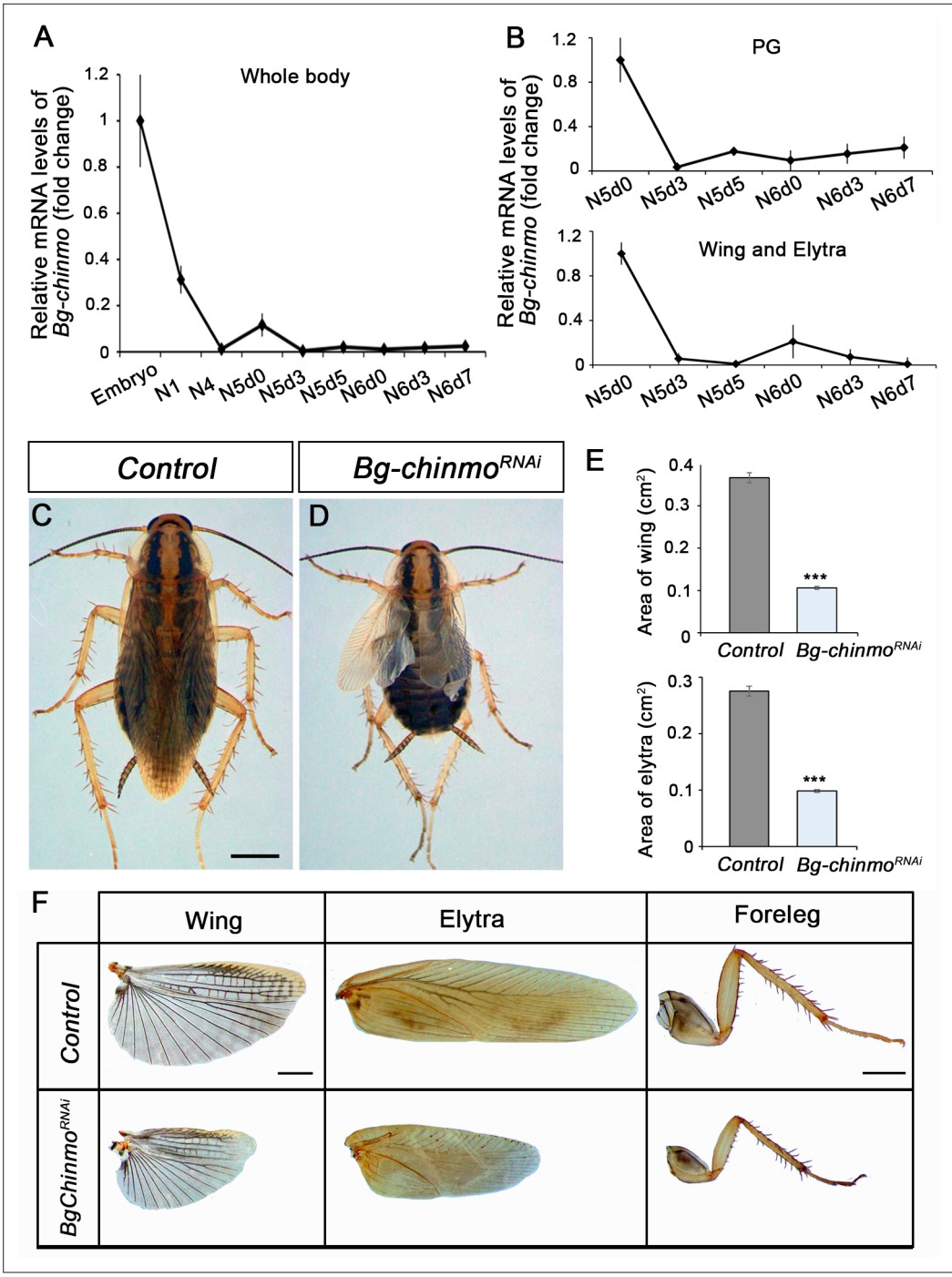

**Figure 9.** Depletion of *chinmo* in *B. germanica* promotes premature adulthood. (**A–B**) *Bg-chinmo* mRNA levels measured by quantitative real-time reverse transcriptase polymerase chain reaction (qRT-PCR) from embryo to the last nymphal stage (**N6**) in whole body (**A**), and prothoracic gland (PG), and wings and elytra (**B**). Transcript abundance values were normalised against the *Rpl32* transcript. Fold changes were relative to the expression of embryo (for whole body) or N5d0 (for PG and wings and elytra), arbitrarily set to 1. Error bars indicate the SEM (n = 3–5). (**C–D**) Newly moulted N4 nymphs of *B. germanica* were injected with *dsMock* (*Control*) or *dschinmo* (*Bg-chinmo^RNAi*) and left until adulthood. (**C**) Dorsal view of adult Control, and (**B**) premature adult *Bg-chinmo^RNAi*. (**E**) Quantification of wing and elytra areas (cm²) of adult Control and *Bg-chinmo^RNAi* premature adults. Error bars indicate the SEM (n = 4–6). Statistical significance was calculated using t-test (***p ≤ 0.001). (**F**) Control and *Bg-chinmo^RNAi* wing, elytra and foreleg of newly emerged adult of *B. germanica*. The scale bar represents 2 mm.

The online version of this article includes the following source data for figure 9:

**Source data 1.** Numerical data for *Figure 9A and D*.

injection of dsRNAs into newly emerged N4 instar. Specimens injected with dsMock were used as negative controls (Control animals). Importantly, whereas Control larvae underwent two nymphal molts before initiating metamorphosis at the end of the N6 stage, 43% of *Bg-chinmo^RNAi* animals underwent only one nymphal molt before molting to an early adult after the N5 stage (*Figure 9B–E*). Precocious *Bg-chinmo*-depleted adults were smaller than control counterparts as they skipped a nymphal stage. However, they presented all the external characteristics of an adult, namely functional hind- and fore-wings, adult legs (*Figure 9F*), adult cerci, mature genitalia, and adult-specific cuticle pigmentation. Altogether, these results suggest that the role of Chinmo as juvenile specifier seems to be conserved in hemimetabolous insects, thereby indicating that its developmental function precedes the hemimetabolous-holometabolous split.

In summary, we identified Chinmo as a new member of the metamorphic gene network acting as a general larval specifier, as recently proposed by *Truman and Riddiford, 2022*. Together with a number of previous reports (reviewed in *Martín et al., 2021*), our results show that the temporal expression of Chinmo, Br-C, and E93 determines the tissue acquisition of gradual differentiation features from the juvenile to the adult to generate the distinct organs. Whereas Chinmo maintains cells in an undifferentiated state, Br-C and E93 induce progressively the differentiation program. This effect has already been shown in the central nervous system where early-born neurons are characterised by the expression of Chinmo, whereas smaller late-born neurons are marked by expression of *Br-C* (*Maurange et al., 2008*). Similarly, the *chinmo*-to-*Br-C* transition in *Drosophila* has been associated with the loss of the regenerative potential of imaginal cells (*Narbonne-Reveau and Maurange, 2019*). The fact that Br-C and E93 act as a tumour suppressor in an overproliferative background supports this idea.

Finally, we found that the role of *chinmo* as larval specifier is conserved in hemimetabolous insects. Since hemimetabolous insects do not undergo the intermediate pupal stage, the transition from juvenile to adult, therefore, relies mainly on the shift from Chinmo to E93 during the last nymphal stage, with Kr-h1 also involved in preventing metamorphosis through the repression of *E93* (*Ureña et al., 2016*). However, although in hemimetabolans insects depletion of *Br-C* does not impair metamorphosis, Br-C does regulate the progressive growth of the metamorphic tissues, such as the wing pads and the dorsal thorax (*Erezyilmaz et al., 2006*; *Fernandez-Nicolas et al., 2022*; *Huang et al., 2013*; *Konopova et al., 2011*). This observation suggests that a premetamorphic role of Br-C was already present in hemimetabolous insects, acquiring an additional metamorphic role as pupal specifier during the evolution of holometabolous insects.

## Materials and methods

### Fly strains

All fly stocks were reared at 25°C on standard flour/agar *Drosophila* media. The *Gal4/UAS* system was used to drive the expression of transgenes at 29°C. *Gal4/Gal80^ts* system was used for conditional activation. In these experiments, crosses were kept at 18 until L2 or L3-late molt and then shifted to 29°C for conditional induction. The following strains used in this study were provided by the Bloomington *Drosophila* Stock Center (BDSC): *fkh-Gal4* (#78060); *Act-Gal4* (#3954); *Tub-Gal80^ts* (#7016), *UAS-chinmo^RNAi* (#26777); *UAS-chinmo* (#50740); *UAS-Br-C^RNAi* (#51378); *UAS-p35* (#5072); *UAS-myr-mRFP* (#7118); and *UAS-mCD8::GFP* (#32186) were used to follow the GAL4 driver activity. *trh-Gal4* (*Kondo and Hayashi, 2013*) and *ey-Gal4* (#8227) were used to drive the expression of distinct constructs in the trachea and eye disc, respectively. A recombinant stock containing UAS-lgl-RNAi[51247] and UAS-lgl-RNAi[51249] (third chromosome) was used to generate the tumours (*Daniel et al., 2018*). *nub-Gal4* (*Calleja et al., 1996*), *esg-Gal4, UAS-GFP* (*Jiang et al., 2009*), and *ci-Gal4* (*Croker et al., 2006*) were used to drive the expression of different constructs in the wing disc. Crosses to *CantonS* line were used as control.

### B. germanica

Specimens of *B. germanica* were obtained from a colony reared in the dark at 30 ± 1°C and 60–70% relative humidity. Cockroaches undergo hemimetabolous development, where growth and maturation take place gradually and simultaneously during a series of nymphal instars. In our rearing conditions, *B. germanica* undergoes six nymphal instars (N1–N6) before molting into the adult. All dissections and tissue sampling were carried out on carbon dioxide anesthetised specimens.

## Immunohistochemistry

For fluorescent imaging, salivary glands, wing discs from different juvenile stages and pupal wings were dissected in 1× phosphate-buffered saline (PBS) and fixed in 4% formaldehyde for 20 min at RT. The tissues were rinsed in 0.1% Triton X-100 (PBST) or 0.3% PBST in pupal wings for 1 hr and incubated at 4°C with primary antibodies diluted in PBST overnight. After incubation with primary antibodies, the tissues were washed with PBST (3×10 min washes) and incubated with adequate combinations of secondary antibodies (Alexa Conjugated dyes 488, 555, 647, Life Technologies, 1:500) for 2 hr at RT, followed by 3×10 min washes with PBST, and then rinsed with PBS before mounting in Vectashield with DAPI (Vector Laboratories, H1200) for image acquisition. The following primary antibodies were used at indicated dilution: rat anti-Chinmo (1:500, N, Sokol), mouse anti-Cut (1:200, Developmental Studies Hybridoma Bank (DSHB) #2B10), mouse anti-Wg (1:200, DSHB #4D4), mouse anti Br-C core (1:250 DSHB #25E9.D7), rabbit anti-cleaved Dcp-1 (1:100, Cell Signaling #9578), and rabbit anti-E93 (1:50, this work).

## Antibody generation

A peptide corresponding to the 23 residues (GRRAYSEEDLSRALQDVVANKL) of E93 was coupled to KLH and was injected into rabbits. Polyclonal antisera were affinity-purified and were found to be specific for E93, by western blotting and by immunofluorescence.

## RNA extraction and quantitative real-time reverse transcriptase polymerase chain reaction

Total RNA was isolated with the GenElute Mammalian Total RNA kit (Sigma), DNAse treated (Promega) and reverse transcribed with Superscript II reverse transcriptase (Invitrogen) and random hexamers (Promega). In the case of *Drosophila*, cDNA was obtained from whole larvae (CantonS) or L3 wandering salivary glands. *B. germanica* cDNAs were obtained from whole nymphs or wings and prothoracic gland (PG) of different juvenile instars. All the samples were collected from females except in the case of L1 and L2 tissues. Relative transcripts levels were determined by real-time polymerase chain reaction (PCR) (quantitative PCR [qPCR]), using iTaq Universal SYBR Green Supermix (Bio-Rad). To standardise the qPCR inputs, a master mix that contained iTaq Universal SYBR Green PCR Supermix and forward and reverse primers was prepared (final concentration: 100 nM/qPCR). The qPCR experiments were conducted with the same quantity of tissue equivalent input for all treatments and each sample was run in duplicate using 2 µl of cDNA per reaction. All the samples were analyzed on the iCycler iQ Real Time PCR Detection System (Bio-Rad). RNA expression was calculated in relation to the expression of *DmRpl32* or *BgActin5C*. Primers sequences for qPCR analyses were (*Duan et al., 2020*):

*Dm-chinmo*-F: 5' AGTTCTGCCTCAAATGGAACAG'3
*Dm-chinmo*-R: 5' CGCAGGATAATATGACATCGGC'3
*Dm-Sgs1*- F: 5'CCCAATCCCGTGTGGCCCTG'3
*Dm-Sgs1*- R: 5' GTGATGGCAACGGCGGTGGT'3
*Dm-Sgs3*- F: 5' TGCTACCGCCCTAGCGAGCA'3
*Dm-Sgs3*- R: 5' GTGCACGGAGGTTGCGTGGT'3
*Dm-Sgs4*- F: 5' ACGCATCAAGCGACACCGCA'3
*Dm-Sgs4*- R: 5'TCCTCCACCGCCCGATTCGT'3
*Dm-Sgs7*- F: 5' CGCAGTCACCATCATCGCTTGC'3
*Dm-Sgs7*- R: 5'ACAGCCCGTGCAGGCCTTTC'3
*Dm-Sgs8*-F: 5' AGCTGCTCGTTGTCGCCGTC'3
*Dm-Sgs8*-R: 5' GCGGAACACCCAGGACACGG'3
*Dm-ng1* and *Dm*ng2-F: 5'- CACTATAAGCGAAAGGTCAAGAG-3'
*Dm-ng1*-R: 5'- TGGATCTTTCATTCATCGGATCT-3'
*Dm-ng2*-R: 5'- TCTTCCGATCTGCGGTTTCTAC-3'
*Dm-ng3*-F: 5'- GATACGACATATCTATAGGCACAA-3'
*Dm-ng3*-R: 5'- CACTGCTGCTACTGCTGCTACT-3'
*Dm-RpL32*-F: 5'CAAGAAGTTCCTGGTGCACAA'3
*Dm-RpL32*-R: 5'AAACGCGGTTCTGCATGAG'3

*Bg-Chinmo-F*: 5' CAGCACCACTATGTCCAAGTG'3
*Bg-Chinmo-R*: 5' CAGGAAACTGGAGAGGCTTTC'3
*Bg-Actin5C-F*: 5'-AGCTTCCTGATGGTCAGGTGA-3'
*Bg-Actin5C-R*: 5'-TGTCGGCAATTCCAGGGTACATGGT-3'

## RNA interference

RNA interference (RNAi) in vivo in nymphs was performed as previously described (*Cruz et al., 2007*; *Martín et al., 2006*). A dose of 1 µl (4–8 µg/µl) of the dsRNA solution was injected into the abdomen of newly antepenultimate (N4d0) instar nymphs, and left until analysed. To promote the RNAi effect, the same dose of dsRNAs was reapplied to all treated animals after 3 days (N4d3) from the first injection. Control dsRNA consisted of a non-coding sequence from the pSTBlue-1 vector (dsControl). The primers used to generate templates via PCR for transcription of the dsRNA were:

*Bg-chinmo-F*: 5'CAGCACCACTATGTCCAAGTG'3
*Bg-chinmo-R*: 5'GAGTCCTGCATGGCTTCGGA'3

## Imaging acquisition and analysis

Images were obtained with the Leica TCS SP5 and the Zeiss LSM880 and LSM780 confocal microscopes. The same imaging acquisition parameters were used for all the comparative analyses. Images were processed with the Imaris Software (Oxford Instruments), Fiji, or Photoshop CS4 (Adobe). For DNA quantification and nuclear size of salivary glands, DNA staining intensity in the salivary glands and tracheal cells was obtained from z stacked images every 0.25 µm of DAPI-stained L3 larvae. Image analysis was performed using Fiji. For the volumetric calculation of the wing pouch region in Imaris software (Oxford Instruments), the regions of interest were selected based on the RFP fluorescence (induced by *nub*-Gal4) in confocal stacks that included the whole disc. The surface function of Imaris was used to segment the wing pouch and the surface volume was calculated by the software. Adult flies, nymphal parts, and adult cockroach images were acquired using AxioImager.Z1 (ApoTome 213 System, Zeiss) microscope, and images were subsequently processed using Photoshop CS4 (Adobe).

## Statistical analysis

Statistical analysis and graphical representations were performed in GraphPad Prism 9 software. All experiments were performed with at least three biological replicates. Two-tailed Student's test and Welch's ANOVA followed by Dunnett's T3 post hoc tests were used to determine significant differences.

## Acknowledgements

The ICTS 'NANOBIOSIS', and particularly the Custom Antibody Service (CAbS, IQAC-CSIC, CIBER-BBN), is acknowledged for the assistance and support related to the E93 antibody used in this work. We thank Josefa Cruz for technical support. This project is supported by grants PGC2018-098427-B-I00 and PID2021-125661NB-I00 to DM and XF-M and grant PGC2018-094254-B-I00 and PID2021-123392NB-I00 to JC funded by MCIN/AEI/10.13039/501100011033 and by grant 2017-SGR 1030 to DM and XF-M funded by the Secretaria d'Universitats i Recerca del Departament d'Economia i Coneixement de la Generalitat de Catalunya and through BIST to JC. The research has also benefited from ERDF 'A way of making Europe to JC, DM and XF-M. SC is a recipient of a Juan de la Cierva FJC2019-041549-I contract from the MCIN.

## Additional information

### Funding

| Funder | Grant reference number | Author |
| --- | --- | --- |
| Ministerio de Ciencia e Innovación | PGC2018-098427-B-I00 | David Martín Xavier Franch-Marro |

| Funder | Grant reference number | Author |
| --- | --- | --- |
| Ministerio de Ciencia e Innovación | PID2021-125661NB-100 | David Martín Xavier Franch-Marro |
| Ministerio de Ciencia e Innovación | PGC2018-094254-B-100 | Jordi Casanova |
| Agència de Gestió d'Ajuts Universitaris i de Recerca | 2017-SGR 1030 | David Martín Xavier Franch-Marro |
| Ministerio de Ciencia e Innovación | PID2021-123392NB-I00 | Jordi Casanova |

The funders had no role in study design, data collection and interpretation, or the decision to submit the work for publication.

## Author contributions
Sílvia Chafino, Conceptualization, Data curation, Formal analysis, Validation, Investigation, Visualization, Methodology, Writing – review and editing; Panagiotis Giannios, Conceptualization, Formal analysis, Validation, Investigation, Visualization, Methodology, Writing – review and editing; Jordi Casanova, Conceptualization, Formal analysis, Supervision, Funding acquisition, Validation, Investigation, Visualization, Project administration, Writing – review and editing; David Martín, Conceptualization, Formal analysis, Supervision, Funding acquisition, Validation, Visualization, Project administration, Writing – review and editing; Xavier Franch-Marro, Conceptualization, Formal analysis, Supervision, Funding acquisition, Validation, Visualization, Writing – original draft, Project administration, Writing – review and editing

## Author ORCIDs
Sílvia Chafino http://orcid.org/0000-0001-9679-0622
Panagiotis Giannios http://orcid.org/0000-0002-7881-1431
Jordi Casanova http://orcid.org/0000-0001-6121-8589
David Martín http://orcid.org/0000-0002-9784-647X
Xavier Franch-Marro http://orcid.org/0000-0002-7465-6729

## Decision letter and Author response
Decision letter https://doi.org/10.7554/eLife.84648.sa1
Author response https://doi.org/10.7554/eLife.84648.sa2

# Additional files

## Supplementary files
• MDAR checklist

## Data availability
All data generated or analysed during this study are included in the manuscript and supporting file. Source Data files have been provided for Figures 1 A and E (Figure1- SourceData1), Figure 2B, C, D and E (Figure2-SourceData1), Figure 5 B, C , D, E and F (Figure5-SourceData1), Figure 8G (Figure8-SourceData1), Figure 9 A and D (Figure9-SourceData1) and Figure2-figure suplement 1 D, E and F (Figure2-figure suplement 1-SourceData1).

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
