## [Editor Report]

This important study demonstrates that the transcription factor Chinmo is a master regulator that maintains larval growth and development as part of the metamorphic gene network in *Drosophila*. Chinmo does so in part by regulating Broad expression in imaginal tissues (e.g. eye and wing discs) and in a Broad-independent manner in other larval tissues such as the salivary glands and larval trachea. Finally, the authors demonstrate that the role of Chinmo in promoting larval development is conserved between holometabolous insects and hemimetabolous insects, which lack a pupal stage. The data were collected and analyzed using solid and validated methodology and will be of interest to a broad audience including those interested in development and evolution.

---

## [Decision Letter]

**Decision letter after peer review:**

Thank you for submitting your article "Antagonistic role of the BTB-zinc finger transcription factors Chinmo and Broad in the juvenile/pupal transition and in growth control" for consideration by *eLife*. Your article has been reviewed by 2 peer reviewers, and the evaluation has been overseen by a Reviewing Editor and Claude Desplan as the Senior Editor. The following individual involved in the review of your submission has agreed to reveal their identity: James W Truman (Reviewer #2).

Essential revisions and clarifications of reviewer feedback

1. Please test the relationship between Chinmo and Broad in imaginal/larval tissues (in addition to the wing disc and salivary glands, which are provided in the manuscript). The reviewers felt this would strengthen the authors' claim that Chinmo promotes larval development in a Br-C-dependent manner in imaginal tissues and a Br-C-independent manner in other larval tissues.

2. Related to point 2 raised by Reviewer #2, following the consultation session, the reviewers agreed that the role of ecdysone and EcR in the transition from Chinmo to Broad is of obvious interest, but that the manuscript stands well without this additional information. While it would be interesting to know if larval cells show a similar transition time and dependence on EcR (perhaps by using an EcR-DN under conditional GAL-80 control) this information does not impact the main message of the manuscript and is therefore not essential.

3. Regarding sex-specific regulation of Chinmo (point 3 raised by reviewer#2), following the consultation session, the reviewers have asked that the figure legend and methods relating to Figure 1 be revised to clarify which sex was used throughout the study.

4. There are some discrepancies with the model that Chinmo promotes larval development and represses the larval-to-pupal transition in larval tissues independently of Broad (see specific reviewer comments) since salivary glands lacking chinmo would be expected to express higher rather than lower levels of sgs genes. This discrepancy may relate to the reduced size of the tissue. It is therefore possible that Chinmo may feed into two parallel pathways, one to regulate growth independently of Broad and the other involving Broad to regulate premetamorphic changes.

- Please examine early larval salivary gland proteins such as ng-1 to -3 that are expressed in salivary glands before the critical weight. Also, it would be interesting if the appearance of the SGS proteins after chinmo knock-down (Figure 5E) is abolished by simultaneous knock-down of broad.

5. Please amend the text to include further consideration of the role of Broad in hemimetabolous insects in light of other publications (see comments from reviewer #3 for more details).

*Reviewer #1 (Recommendations for the authors):*

Please accept these manuscript comments, which include experimental suggestions as well as minor spelling errors:

1. In the legend for Figure 1E, the larval stage isolated at the time of qPCR should be reported.

2. In Figure 1F, the staining of Br-C and Chinmo in L3W wing discs should be repeated with proper controls included: control wing discs, UAS-chinmoRNAi with an additional "control" UAS-line such as UAS-laczRNAi. The latter should be included since the stated conclusion of this experiment is that wing disc growth requires the absence of Br-C until the L3W stage. The restoration of normal growth in the experiment shown could be a result of attenuated Chinmo depletion due to the presence of a 3rd UAS construct (there is already a UAS-GFP in the background).

3. For similar reasons stated above, the following experiments/figures should be shown with proper controls:

– Figure 5A (single knockdowns of Br-C or chinmo, each with UAS control);

– Figure 5F (control wing disc staining, as well as UAS-chinmoRNAi with a UAS control).

4. Control stainings should be included for:

– Br-C and Ct staining in control wing discs are not shown in any figures and should be included in experimental controls for each figure where this staining appears (e.g., Fig6);

– Same comment for E93 in Fig7.

5. Stage of animals used to assess SG gene expression in Figure 2E should be reported.

6. In Figure 4, unique subfigure letters for each genotype would aid analysis/reading.

7. Unique figure letters for each graph in 9A would aid reader analysis. Also, the "PG" and "wing and elytra" graphs should each include a y-axis label.

8. In the text description of the esgGal4 driver (lines 150-151), I suggest revising it to the following: "…we knocked down this factor in the pouch region of wing imaginal discs…"

9. In line 181, there is a minor typo. Would revise it to "As a consequence, precocious upregulation of Br-C blocked development in both tissues, phenocopying the loss of chinmo.

10. In lines 205-213, literature evidence is presented to bolster the notion that MGN factors can exert different functions depending on the tissue, as they demonstrate Chinmo acting in both Br-C-dependent and independent manners to promote larval development. I'm not sure that I logically follow this argument and I don't think it strengthens the previous statement. If it is possible to clarify the argument then I support that, but otherwise, I would suggest removing this portion of the text.

11. In line 282, remove the word "also"

12. In lines 315-317, the authors describe phenotypes that are preserved in the absence of Bg-chinmo (Figure 9). I recommend including representative images that show that the mentioned adult structures are preserved in Bg-chinmoRNAi.

13. In line 335, there is a typo in the word "background".

*Reviewer #2 (Recommendations for the authors):*

Overall, the paper is a very nice piece of work. Besides my comments in the Public Review, there are a couple of general issues in the presentation. One is that through the text and figures the authors might use FlyBase terminology for the different genetic constructs: "UAS-chinmo" rather than "UASChinmo", nub-GAL4 rather than nubGal4, and esg-GAL4, UAS-GFP" rather than "esgGal4UASGFP" (ln 359).

A second general issue is that the font for the labels on some of the confocal images is so small that it is unreadable (Figure 4, 5F, 6, 7, and 8A)

---

## [Author Response]

Essential revisions and clarifications of reviewer feedback1. Please test the relationship between Chinmo and Broad in imaginal/larval tissues (in addition to the wing disc and salivary glands, which are provided in the manuscript). The reviewers felt this would strengthen the authors' claim that Chinmo promotes larval development in a Br-C-dependent manner in imaginal tissues and a Br-C-independent manner in other larval tissues.

We have analysed the role of *chinmo* in two additional tissues as suggested by the referees and the editorial board. The larval tracheal system as an additional larval tissue and the eye disc as another imaginal tissue. Confirming the results obtained in the salivary glands and wing discs, we found that *chinm*o promotes development in the eye imaginal disc by repressing precocious expression of Br-C, whereas in the larval trachea, in addition to the Br-C repression, induces tracheal growth. We generated two Figures supplement with the new results and added the data in the main text. (Please see the complete explanation below).

2. Related to point 2 raised by Reviewer #2, following the consultation session, the reviewers agreed that the role of ecdysone and EcR in the transition from Chinmo to Broad is of obvious interest, but that the manuscript stands well without this additional information. While it would be interesting to know if larval cells show a similar transition time and dependence on EcR (perhaps by using an EcR-DN under conditional GAL-80 control) this information does not impact the main message of the manuscript and is therefore not essential.

We agree with the editor that the analysis of ecdysone signalling, although of interest, would not substantially modify the message of the manuscript and, therefore, we did not address it (Please see the complete explanation below).

3. Regarding sex-specific regulation of Chinmo (point 3 raised by reviewer#2), following the consultation session, the reviewers have asked that the figure legend and methods relating to Figure 1 be revised to clarify which sex was used throughout the study.

We have performed the measurements of *chinmo* expression always in females when sex identification was possible, namely in L3. In L1 and L2 larvae, sex identification was not possible in our conditions. Nevertheless, we think that there are evidences that suggest that *chinmo* does not have a significant role in sex difference growth rate during larval development, when precisely male and female growth differences have been reported (Please see the complete explanation below).

4. There are some discrepancies with the model that Chinmo promotes larval development and represses the larval-to-pupal transition in larval tissues independently of Broad (see specific reviewer comments) since salivary glands lacking chinmo would be expected to express higher rather than lower levels of sgs genes. This discrepancy may relate to the reduced size of the tissue. It is therefore possible that Chinmo may feed into two parallel pathways, one to regulate growth independently of Broad and the other involving Broad to regulate premetamorphic changes.- Please examine early larval salivary gland proteins such as ng-1 to -3 that are expressed in salivary glands before the critical weight. Also, it would be interesting if the appearance of the SGS proteins after chinmo knock-down (Figure 5E) is abolished by simultaneous knock-down of broad.

We measured the expression levels of early larval salivary gland protein *ng-1*, *-2* and *-3* genes in *chinmo* and in *chinmo* and *Br-C* depleted animals. As happens to be the case also for *Sgs* genes, the expression levels of those genes were virtually undetectable in both conditions, suggesting that *chinmo* is required for SG growth and maturation. (Please see the complete explanation below).

5. Please amend the text to include further consideration of the role of Broad in hemimetabolous insects in light of other publications (see comments from reviewer #3 for more details).

We clarified the role of Br-C in hemimetabolous insects in the main text as suggested.

Reviewer #1 (Recommendations for the authors):Please accept these manuscript comments, which include experimental suggestions as well as minor spelling errors:1. In the legend for Figure 1E, the larval stage isolated at the time of qPCR should be reported.

This information was already reported in the figure legends of Figure 1: “qPCR was performed from SG dissected at L1 larva”.

2. In Figure 1F, the staining of Br-C and Chinmo in L3W wing discs should be repeated with proper controls included: control wing discs, UAS-chinmoRNAi with an additional "control" UAS-line such as UAS-laczRNAi. The latter should be included since the stated conclusion of this experiment is that wing disc growth requires the absence of Br-C until the L3W stage. The restoration of normal growth in the experiment shown could be a result of attenuated Chinmo depletion due to the presence of a 3rd UAS construct (there is already a UAS-GFP in the background).

We think that the referee refers to Figure 5F and not Figure1F as there is no such panel 1F in the Figure 1. In any case, we understand the concern of the referee regarding the strength of the Gal4 with multiple UAS transgenes. For this reason, we provide a staining of Br-C and Chinmo in the double knockdown in Figure 5F to prove that the downregulation of each gene is indeed effective. The same control of RNAi effectiveness was performed for the SG. We added the staining of Br-C and Chinmo in depleted SGs in a new Figure 5—figure supplement 1. In summary, we believe that measuring the decay of protein levels in each experimental condition is a strong enough evidence of the RNAi efficiency.

3. For similar reasons stated above, the following experiments/figures should be shown with proper controls:– Figure 5A (single knockdowns of Br-C or chinmo, each with UAS control);– Figure 5F (control wing disc staining, as well as UAS-chinmoRNAi with a UAS control).

As in the previous point, we think that staining for Br-C and Chinmo in the SG is a good enough control to show the efficiency of the RNAi. Importantly, the double knockdown of *Br-C* and *chinmo* is not able to rescue the *chinmo* RNAi alone. This observation supports the idea that the double UAS is not reducing the effectiveness of the RNAi, as, otherwise, we should observe a different phenotype. Nevertheless, as we did for the imaginal disc, we measured the expression of Br-C and Chinmo in a double knockdown SGs. Under these conditions we were not able to detect neither Chinmo nor Br-C demonstrating that both RNAis work properly. We added this data in a new Figure 5—figure supplement 1.

For the experiment presented in panel F of Figure 5, we generated a new Figure 5—figure supplement 2 with the staining of Chinmo, Br-C and Wg in *Control* wing discs.

4. Control stainings should be included for:– Br-C and Ct staining in control wing discs are not shown in any figures and should be included in experimental controls for each figure where this staining appears (e.g., Fig6);– Same comment for E93 in Fig7.

Control Ct staining is shown in Figure 3. In Figures 6 and 7 all the experiments were performed with internal controls. We used the cubitus interruptus driver (*ci-Gal4*), which is only expressed in the anterior compartment of the larval disc or pupal wing. Therefore, wild type expression of the different markers is provided in these two figures in the posterior compartment. For this reason, we think that it is redundant and confusing to add the expression of the different markers analysed in a wild type disc in those special conditions.

5. Stage of animals used to assess SG gene expression in Figure 2E should be reported.

SGs were dissected from L3W larva. We have added this information in the new Figure 2 legend.

6. In Figure 4, unique subfigure letters for each genotype would aid analysis/reading.

We labelled each genotype with a letter as suggested.

7. Unique figure letters for each graph in 9A would aid reader analysis. Also, the "PG" and "wing and elytra" graphs should each include a y-axis label.

We modified the figure as suggested.

8. In the text description of the esgGal4 driver (lines 150-151), I suggest revising it to the following: "…we knocked down this factor in the pouch region of wing imaginal discs…"

We modified the sentence as suggested.

9. In line 181, there is a minor typo. Would revise it to "As a consequence, precocious upregulation of Br-C blocked development in both tissues, phenocopying the loss of chinmo.

We modified the sentence as suggested.

10. In lines 205-213, literature evidence is presented to bolster the notion that MGN factors can exert different functions depending on the tissue, as they demonstrate Chinmo acting in both Br-C-dependent and independent manners to promote larval development. I'm not sure that I logically follow this argument and I don't think it strengthens the previous statement. If it is possible to clarify the argument then I support that, but otherwise, I would suggest removing this portion of the text.

We clarified this point in the main text.

11. In line 282, remove the word "also"

We modified the sentence as suggested.

12. In lines 315-317, the authors describe phenotypes that are preserved in the absence of Bg-chinmo (Figure 9). I recommend including representative images that show that the mentioned adult structures are preserved in Bg-chinmoRNAi.

We included the adult structure of a wing, elytrum and leg of *Control* and *Bg-chinmo^RNAi^* animals in a remodelled Figure 9 to better show the premature metamorphosis observed upon *Bg-Chinmo* depletion.

13. In line 335, there is a typo in the word "background".

We fixed the typo mistake.

Reviewer #2 (Recommendations for the authors):Overall, the paper is a very nice piece of work. Besides my comments in the Public Review, there are a couple of general issues in the presentation. One is that through the text and figures the authors might use FlyBase terminology for the different genetic constructs: "UAS-chinmo" rather than "UASChinmo", nub-GAL4 rather than nubGal4, and esg-GAL4, UAS-GFP" rather than "esgGal4UASGFP" (ln 359).A second general issue is that the font for the labels on some of the confocal images is so small that it is unreadable (Figure 4, 5F, 6, 7, and 8A)

We increased the font labels of the Figures to make them more visible.